# Scaling Behaviour and Critical Phase Transitions in Integrated Information Theory

**DOI:** 10.3390/e21121198

**Published:** 2019-12-05

**Authors:** Miguel Aguilera

**Affiliations:** 1IAS-Research Center for Life, Mind and Society, University of the Basque Country, 20018 Donostia, Spain; sci@maguilera.net; 2ISAAC Lab, Aragón Institute of Engineering Research, University of Zaragoza, 50018 Zaragoza, Spain

**Keywords:** Integrated Information Theory, Phi, Ising model, criticality, phase transitions

## Abstract

Integrated Information Theory proposes a measure of conscious activity (Φ), characterised as the irreducibility of a dynamical system to the sum of its components. Due to its computational cost, current versions of the theory (IIT 3.0) are difficult to apply to systems larger than a dozen units, and, in general, it is not well known how integrated information scales as systems grow larger in size. In this article, we propose to study the scaling behaviour of integrated information in a simple model of a critical phase transition: an infinite-range kinetic Ising model. In this model, we assume a homogeneous distribution of couplings to simplify the computation of integrated information. This simplified model allows us to critically review some of the design assumptions behind the measure and connect its properties with well-known phenomena in phase transitions in statistical mechanics. As a result, we point to some aspects of the mathematical definitions of IIT that 3.0 fail to capture critical phase transitions and propose a reformulation of the assumptions made by integrated information measures.

## 1. Introduction

Integrated Information Theory (IIT [1]) was developed to address the problem of consciousness by characterizing its underlying processes in a quantitative manner. It provides a measure of integration, Φ, that quantifies to what extent a dynamical system generates information that is irreducible to the sum of its parts, considered independently. Beyond IIT as a theory of consciousness, Φ has received attention as a general measure of complexity, and different versions of the measure have been applied to capture to what extent the behaviour of a system is both differentiated (displaying diverse local patterns) and integrated (maintaining a global coherence). Furthermore, Φ attempts to capture the level of irreducibility of the causal structures of a system, revealing the boundaries in the organisation of complex dynamical systems (i.e., delimiting the parts of the system that are integrated into a functional unit [2]).

Despite its promising features, IIT is still controversial and it has received different critiques, both to its theoretical and philosophical foundations (e.g., see [3,4]) and the mathematical definitions therein. Some of the latter suggest that Φ might not be well defined and presents a number of problems [5]. Furthermore, there is a myriad of different definitions of Φ, and experimental testing of these competing definitions in small systems shows that their properties radically diverge in some cases [6]. Despite the abundance of critiques and alternative definitions of Φ, it is not clear which is the appropriate direction to settle theoretical differences or test different approaches experimentally. In our view, there is two main obstacles that hinder this endeavour: (1) the difficulty of testing integration measures in well-known dynamical systems where integrated information measures can be evaluated against known properties of the system, and (2) the computational cost of calculating IIT measures, preventing its application beyond small networks.

The first problem is, in general, difficult, as even in simple nonlinear models the relation between the network topology and dynamics is complex, and it is not always clear which topology should yield larger integrated information. Nevertheless, there is a family of models in which the relation between segregation and integration is well characterised: homogeneous systems exhibiting order-disorder critical phase transitions [7]. In these systems, there is a transition in their phase space from a disordered state, in which activity is random and segregated, to an ordered one, where units of the system are strongly coordinated. Just at the boundary separating ordered and disordered dynamics we find criticality, a state where a compromise between coordination and segregation is found [7,8]. Even in simple systems, critical dynamics are dominated by small bursts of local (i.e., segregated) activity, yet display large avalanches of globally coordinated (i.e., integrated) activity. In neural systems, these are generally referred to as “neuronal avalanches”, and experimental evidence suggests that neural dynamics exhibit a degree of concordance with those expected for a system near criticality [9].

Critical phenomena are theoretically characterised for some systems of infinite size, and they can be experimentally identified as divergent tendencies in large, finite systems as they grow in size. This refers us to the second problem, which is the very large computational cost of measuring integrated information. Due to combinatorial explosion, computing Φ is only possible in general for small, discrete systems. In practice, this prevents to measure integrated information in the very large or even infinite systems where critical dynamics can be appreciated. In IIT 3.0 [1], the latest version of the framework of Integrated Information Theory, Φ can only be computed for binary networks composed of up to a dozen units. There is a variety of other measures to compute integrated information [6], and some of them are computationally lighter, but all share these limits to some extent. As most versions of Φ require computing distance measures between probability distributions of a system and finding the minimum information partition (MIP), they present important restrictions in terms of computational cost as a system scales in size. Note that some simplified measures of Φ can be computed analytically for the case of Gaussian distributions (e.g., see [10,11]), but, in this paper, we will focus in the general case of binary variables, without considering the specific case of Gaussian assumptions.

In general, IIT measures have been limited to very small systems, and it is not well understood how integrated information scales with the size or temporal scale of the system. Previous work [12] has analysed how integrated information changes with spatial and temporal coarse graining in small networks, but it is still difficult to connect these results with well-known scaling effects like the ones that take place in critical phase transitions. Luckily, a family of simplified models, generally referred to as Ising models, can capture critical transitions with very simple topologies. Some of the simplest ones present homogeneous architectures that can greatly simplify the calculation of information theoretical quantities of the system, see, e.g., [13]. Using this idea, recent novel work using mean-field approximations [14] has shown that under some assumptions it is possible to compute integrated information in infinite-size networks with some homogeneous properties, showing that integrated information measures diverge at certain critical points. In this work, we extend those results by finding methods for computing integrated information of similar models of finite size. Specifically, we explore how integrated information measures scale with size in large networks, proposing methods for simplifying the computation of distance metrics between probability distributions and the search of the MIP. In doing so, we critically assess some of the assumptions behind IIT measures and propose some modifications to better capture the properties of second-order phase transitions. Specifically, we will explore different aspects of the definition of integrated information: (1) the dynamics and temporal span of integrated information, (2) assumptions for the computation of the cause repertoire, (3) the choice of distance metrics between the Wasserstein distance and the Kullback–Leibler divergence, (4) the effect of considering the environment from a situated perspective, (5) the relation between mechanism-level integration φ and system-level integration Φ and (6) the importance of identifying diverging tendencies in the system.

## 2. Model

To show how integrated information behaves around critical points in Ising models, we describe a (slightly modified) version of IIT 3.0. Then, we introduce a family of homogeneous kinetic Ising models and a series of methods that simplify the computation of integrated information in large networks.

### 2.1. IIT 3.0

We critically revise and adapt the framework of IIT 3.0 [1], which originally computes the integrated information of a subset of elements of a system as follows. For a system of *N* elements with state s at time *t* (We use boldface letters and symbols for vectors and matrices, e.g., $s\left(t\right)={\left(s_{1}\left(t\right),s_{2}\left(t\right),\cdots ,s_{N}\left(t\right)\right)}^{T}$, ${}^{T}$ indicating transposition.), we characterise the input–output relationship of the system elements through its corresponding transition probability function P(s(t+τ)|s(t)), describing the probabilities of the transitions from one state to another for all possible system states. IIT 3.0 requires systems to satisfy the Markov property (i.e., that the distribution of states of a process at time *t* depends only upon the state at time t−τ), and that the current states of elements are conditionally independent, given the past state of the system, i.e., $P\left(s\left(t\right)|s\left(t-\tau \right)\right)=\prod_{i}P\left(s_{i}\left(t+\tau \right)|s\left(t\right)\right)$.

IIT 3.0 computes integrated information in the causal mechanisms of a system by defining two subsets of s(t) and s(t±τ), called the mechanism $M_{t}={\left\{s_{i}\left(t\right)\right\}}_{i\in I_{M_{t}}}$ and the purview $P_{t\pm \tau }={\left\{s_{i}\left(t\pm \tau \right)\right\}}_{i\in I_{P_{t\pm \tau }}}$, to represent the current state of part of the system and how it constrains future or past states. The cause-and-effect repertoires of the system are described, respectively, by the probability distributions $P(P_{t-\tau }|M_{t})$ and $P(P_{t+\tau }|M_{t})$.

The integrated cause–effect information of $M_{t}$ is then defined as the distance between the cause–effect repertoires of a mechanism and the cause–effect repertoires of its minimum information partition (MIP) over the maximally irreducible purview,
(1)$$\begin{array}{cc} \varphi_{\mathrm{cause}}\left(\tau \right) & =\underset{P}{max}\left(\underset{\mathrm{cut}}{min}\left(D_{W}\left(P\left(P_{t-\tau }|M_{t}\right),P_{\mathrm{cut}}\left(P_{t-\tau }|M_{t}\right)\right)\right)\right), \end{array}$$
(2)$$\begin{array}{cc} \varphi_{\mathrm{effect}}\left(\tau \right) & =\underset{P}{max}\left(\underset{\mathrm{cut}}{min}\left(D_{W}\left(P\left(P_{t+\tau }|M_{t}\right),P_{\mathrm{cut}}\left(P_{t+\tau }|M_{t}\right)\right)\right)\right), \end{array}$$
where $D_{W}\left(P,Q\right)$ refers to the Wasserstein distance (also known as Earth Mover’s Distance), used by IIT 3.0 to quantify the statistical distance between probability distributions *P* and *Q*. The subindex cut specifies a bipartition of the mechanism into two halves and $P_{\mathrm{cut}}$ represents the cause or effect probability distribution under such partition,
(3)$$\begin{array}{cc} \mathrm{cut} & =\left\{M_{t}^{1},P_{t\pm \tau }^{1},M_{t}^{2},P_{t\pm \tau }^{2}\right\},\\ P_{\mathrm{cut}}\left(P_{t\pm \tau }|M_{t}\right) & =P\left(P_{t\pm \tau }^{1}|M_{t}^{1}\right)⊗P\left(P_{t\pm \tau }^{2}|M_{t}^{2}\right). \end{array}$$


Here, cut specifies the partition applied over the elements of mechanism M, where $M_{t}^{1},M_{t}^{2}$ design the blocks of a bipartition of the mechanism at the current state at time *t*, $M_{t}$; $P_{t\pm \tau }^{1},P_{t\pm \tau }^{2}$ refer to the blocks of a bipartition (not necessarily the same) of the present or past units $P_{t\pm \tau }$. Figure 1B represents the partition $M_{1}=\left\{s_{1}\left(t\right),s_{2}\left(t\right)\right\},M_{2}=\left\{s_{3}\left(t\right)\right\},P_{1}=\left\{s_{1}\left(t+1\right),s_{2}\left(t+1\right),s_{3}\left(t+1\right)\right\},P_{2}=\left\{\right\}$. The interaction between the partitioned systems (⊗ operator) is defined by injecting uniform random noise in the partitioned connections when defining the transition probability matrix P(s(t±τ)|s(t)).

The integrated information of the mechanism $M_{t}$ with a time span τ, φ(τ), is the minimum of its corresponding integrated cause and effect information,
(4)$$\varphi =\min(\varphi_{\mathrm{cause}},\varphi_{\mathrm{effect}}).$$


The integrated information of the entire system, Φ(τ), is then defined as the distance between the cause–effect structure of the system and the cause–effect structure of its minimum information partition, eliminating constraints from one part of the system to the rest:(5)$$\Phi \left(\tau \right)=\underset{\mathrm{cut}}{min}\;D_{W}^{*}\left(C\left(\tau \right),C_{\mathrm{cut}}\left(\tau \right)\right),$$
where C(τ) stands for a “constellation of concepts”, which is constructed from the set of points with position {P(s(t−τ)|s(t)),P(s(t+τ)|s(t))} and value φ(τ) corresponding to all the mechanisms in the system. Similarly, $C_{\mathrm{cut}}\left(\tau \right)$ stands for a constellation of a system in which a new unidirectional partition has been applied, injecting noise in the partitioned connections as in previous case (note that now φ is computed applying two different partitions). In this case, a especial distance measure is used (we label it as $D_{W}^{*}$), which is a modified he extended version of the Wasserstein distance that measures the minimal transport cost of transforming one constellation into another [1] (Text S2), in which φ(τ) are the values to be transported and the Wasserstein distance between {P(s(t−τ)|s(t)),P(s(t+τ)|s(t))} in the original system and under the partition is the distance these values have to be transported. Finally, if the system is a subset of elements of a larger system, all elements outside the system are considered as part of the environment and are conditioned on their current state throughout the causal analysis. Similarly, when evaluating a mechanism, all elements inside a mechanism (where φ is analysed) but outside the system (where Φ is determined) are considered as uniform, independent noise sources. Further details of the steps described here can be found in [1].

#### Working Assumptions

In order to compute integrated information in large systems, we modify some aspects of the theory. In IIT 3.0, an integrated information of a mechanism φ is evaluated for a particular mechanism $M_{t}$ and a purview $P_{t\pm \tau }$. Here, for simplicity, we assume that the purview always includes the same units as the mechanism (although we allow them to be partitioned differently, see, e.g., Figure 1B). Allowing more options for the purview could make a big difference in some systems; although, in the homogeneous systems tested here, the differences are small. Also, the distance for computing integrated information is measured for the distance of all elements of the system, not only the elements contained in the purview. In IIT 3.0 only elements in the purview are affected by a partition. In our modified version of the measure, in some cases (τ>1, see Section 3.1) the outside of the purview can change as well, thus capturing these changes offers a better characterisation of the effects of partitions.

Moreover, as we assume a homogeneous architecture, in some cases mechanism integration φ and system-level integration Φ have a similarly diverging behaviour (as we explore in Section 3.5). Thus, for simplicity, in most cases we compute only the integrated information φ of a mechanism comprising the system of interest.

This homogeneous architecture also allows us to assume that under some conditions (systems with possible couplings and near the thermodynamic limit) the MIP is always either a partition that cuts a single node from the mechanism of the system or a cut that separates entire regions in different partitions (see Section B.3). This assumption will reduce drastically the computational cost of calculating integrated information.

Other assumptions will be studied in different sections of the article. In Table A1, these different assumptions are described and in Table A2 it is indicated if they are used by IIT 3.0 and if they are applied for obtaining the results of the different figures of Section 3.

### 2.2. Kinetic Ising Model with Homogeneous Regions

We define a general model capturing causal interactions between variables. Aiming for generality, we use the least structured statistical model defining causal correlations between pairs of units from one time step to the next [15]. We study a kinetic Ising model where *N* binary spins $s_{i}\in \left\{-1,+1\right\}$ evolve in discrete time, with synchronous parallel dynamics (Figure 1A). Given the configuration of spins at time *t*, $s\left(t\right)=\{s_{1}\left(t\right),\cdots ,s_{N}\left(t\right)\}$, the spins $s_{i}\left(t\right)$ are independent random variables drawn from the distribution:(6)$$\begin{array}{cc} P(s_{i}\left(t\right)|s\left(t-1\right)) & =\frac{e^{\beta s_{i}\left(t\right)h_{i}\left(t\right)}}{2\cosh(\beta h_{i}\left(t\right))}, \end{array}$$(7)$$\begin{array}{cc} h_{i}\left(t\right) & =H_{i}+\sum_{j}J_{ij}s_{j}\left(t-1\right). \end{array}$$

The parameters H and J represent the local fields at each spin and the couplings between pairs of spins, and β is the inverse temperature of the model. Without loss of generality, we assume that β=1.

In general, computing the probability distributions P(s(t)) of a kinetic Ising model is a difficult task, as it requires computing over all previous trajectories. In general, P(s(t+τ)|s(t)) is computed recursively applying the equation
(8)$$P\left(s\left(t+\tau \right)|s\left(t\right)\right)=\sum_{s(t+\tau -1)}P\left(s\left(t+\tau \right)|s\left(t+\tau -1\right)\right)P\left(s\left(t+\tau -1\right)|s\left(t\right)\right).$$


The cost of calculating this equation grows exponentially with size of the system, with a computational cost of $O(2^{2N})$.

This computation can be simplified for certain architectures of the model. We divide the system into different regions, and assume that the coupling values $J_{ij}$ are positive and homogeneous for each intra- or inter-region connections $J_{ij}=\frac{1}{N_{V}}J_{UV}$, where U and V are regions of the system with sizes $N_{U},N_{V}$ and i∈U,j∈V. Also, for simplicity we assume that H=0.

When the system is divided in homogeneous regions, the calculation of the probability distribution of the system is simplified to computing the number of active units for each region $S_{U}\left(t\right)=\sum_{i\in U}\left(1+s_{i}\left(t\right)\right)/2$. With this, we simplify the transition probability matrix to
(9)$$\begin{array}{cc} P(s_{i}\left(t\right)|S\left(t-1\right)) & =\frac{e^{\beta s_{i}\left(t\right)h_{i}\left(t\right)}}{2\cosh(\beta h_{i}\left(t\right))},\;h_{i}\left(t\right)=H_{i}+\sum_{V}J_{UV}S_{V}\left(t-1\right), \end{array}$$
(10)P(SU(t)|S(t−1))=P(si(t)=1|S(t−1))SU(t)P(si(t)=0|S(t−1))(NU−SU(t))NUSU(t).


Having then that
(11)$$P\left(S\left(t+\tau \right)|S\left(t\right)\right)=\sum_{S(t+\tau -1)}\prod_{U}P\left(S_{U}\left(t+\tau \right)|S\left(t+\tau -1\right)\right)P\left(S\left(t+\tau -1\right)|S\left(t\right)\right).$$


Now the cost is reduced to $O(\prod_{U}{\left(N_{U}+1\right)}^{2})$, which, for a limited number of regions, makes the computation much lighter.

Interestingly, as shown in [14], if the size of the regions tends to infinity, then the behaviour of a region U is described simply by the mean field activity of its units, $m_{U}\left(t\right)=\frac{1}{N_{U}}\sum_{j\in U}s_{j}\left(t\right)$, and its evolution becomes deterministic, having that the behaviour of any unit *i* belonging to a region V by the value of the input field of the region $h_{V}$:(12)$$\begin{array}{cc} P(S_{U}\left(t\right)|S\left(t-1\right)) & =\delta (S_{U}\left(t\right)-N_{U}\frac{1+m_{U}\left(t\right)}{2}), \end{array}$$(13)$$\begin{array}{cc} m_{U}\left(t\right) & =\tanh\sum_{V}J_{UV}m_{V}\left(t-1\right), \end{array}$$
where δ(x) is the Kronecker delta function and $m_{U}\left(t-1\right)$ is the mean field of region U(t−1).

### 2.3. Integrated Information in the Kinetic Ising Model with Homogeneous Regions

Describing an Ising model with homogeneous regions simplifies the computation of integrated information in two important ways: by reducing the costs of finding the minimum information partition and computing statistical distances between distributions.

As the connectivity of the system is homogeneous for all nodes in the same region, in Section B.3 we show that, near the thermodynamic limit and for the case of only positive couplings, the MIP is always a partition that either (a) isolates only one unit from one of the regions of the system, or (b) separates entire regions such that all elements of a region in the current or the future (or past) states always belong to the same partition. Also, in case (a), the partition that isolates a single unit in time *t* always has a smallest value of φ than the partition isolating a node at time t±τ, as partitioning the posterior distribution corresponds to a larger distance between probability distributions (see Section B.3). We tested both cases (a) and (b) and found that in all the examples exemplified in this article, as all couplings are in a similar order of magnitude, the MIP always is as in case (a), so the MIP always cuts only a node at time *t* from one region of the system (see, e.g., Figure 1B). In this work, we also compute the value of Φ for a homogeneous system with just one region. In this case, as there is only one region, the MIP at the system level is also a partition that isolates only one node, as this is the intervention yielding a minimal distance (see Section B.4).

In systems with finite size, the evolution of the probability distribution of the activity at the regions of the system is calculated using Equation (Equation 11). From there, Equations (Equation 1) and (2) can be computed. For large systems, it becomes unfeasible to compute the Wasserstein distance between distributions due to the combinatorial explosion of states. Nevertheless, when regions are homogeneous this computation is greatly simplified if the number of regions is not too large. As all the units within a region are statistically identical, in terms of Wasserstein distances it is equivalent to work with aggregate variables representing the sum of the units of a region (see Section B.1):(14)$$\begin{array}{c} \varphi_{M}^{\mathrm{cut}}\left(\tau \right)=D_{W}(P\left(s\left(\tau_{0}+\tau \right)|s\left(\tau_{0}\right)\right)\left|\right|P_{\mathrm{cut}}\left(s\left(\tau_{0}+\tau \right)|s\left(\tau_{0}\right)\right))\\ =D_{W}((P\left(S\left(\tau_{0}+\tau \right)|s\left(\tau_{0}\right)\right)\left|\right|P_{\mathrm{cut}}\left(S\left(\tau_{0}+\tau \right)|s\left(\tau_{0}\right)\right)). \end{array}$$

This equivalence is possible because the transport cost between two states s(t) and $s^{*}\left(t\right)$ is defined as $\frac{1}{2}\sum_{i}\left|s_{i}\left(t\right)-s_{i}^{*}\left(t\right)\right|$. As the Wasserstein distance always chooses minimal transport costs, then the cost between states S(t), $S^{*}\left(t\right)$ is defined as $|S\left(t\right)-S^{*}\left(t\right)|$. If instead of the Wasserstein distance we use the Kullback–Leibler divergence, then $D_{KL}\left(s\left(t\right)|\right|s^{*}\left(t\right))=D_{KL}\left(S\left(t\right)|\right|S^{*}\left(t\right))$ (see Appendix C).

Finally, when the partition only affects one node of the mechanism of the system at region V (see, e.g., Figure 1B), the computation of $P_{\mathrm{cut}}$ is performed by transforming the transfer probability matrix as
(15)$$\begin{array}{cc} P_{\mathrm{cut}}\left(S_{U}\left(t\right)|S\left(t-1\right)\right)= & \frac{1}{2}(P\left(S_{U}\left(t\right)|S\left(t-1\right)\right)\\ & +\left(1-\frac{S_{V}\left(t\right)}{N_{V}}\right)P\left(S_{U}\left(t\right)|S_{V}\left(t-1\right)+1,S_{\bar{V}}\left(t-1\right)\right)\\ & +\frac{S_{V}\left(t\right)}{N_{V}}P\left(S_{U}\left(t\right)|S_{V}\left(t-1\right)-1,S_{\bar{V}}\left(t-1\right)\right)), \end{array}$$
where $\bar{V}$ is the complement set to {V}. The origin of this expression is that injecting uniform noise to a single unit of region V can have three possible outputs: leaving the system as it was ($\frac{1}{2}$ chance), adding one to the value of $S_{V}$ ($\frac{1}{2}\left(1-\frac{S_{V}\left(t\right)}{N_{V}}\right)$ chance) or subtracting one to the value of $S_{V}$ ($\frac{1}{2}\frac{S_{V}\left(t\right)}{N_{V}}$ chance). The linear combination of these three cases yields the final transfer probability matrix.

When computing integrated information of the system, Φ, Equation (Equation 5) computes the distance between concepts (i.e., values of integration of the mechanisms of a system) of the original system and the system under a unidirectional partition. Mechanisms affected by the partition will always have a value of φ=0 (and are transported to a residual, “null” concept located at the unconstrained distribution, see [1] (Text S2)). In our example, we find that these concepts contribute the most to the value of Φ (see Section 3.5).

## 3. Results

Criticisms concerning the definition of integrated information measures have addressed a variety of topics, e.g., the existence of “trivially non-conscious” systems, composed of units distributed in relatively simple arrangements yielding arbitrarily large values of Φ, or the absence of a canonical metric of the probability distribution space [5]. There are also some aspects that are not very well understood as the dependence of Φ with the scale or graining of a system, or the differences and dependencies between cause and effect repertoires. Here, we explore some of these aspects and explore possible reformulations of the measures concerning how current measures behave around critical phase transitions. We introduce as a reference the behaviour of the kinetic Ising model with homogeneous regions of infinite size. As described in Equation (Equation 12), behaviour in the thermodynamic limit can be described by the evolution of the mean firing rates. Also, computing the derivative of the mean firing rates with infinite size (which will be used to compute the distances between distributions) is straightforward:(16)$$\begin{array}{cc} \frac{\partial m_{U}\left(t\right)}{\partial J_{UV}} & =\left(1-m_{U}^{2}\left(t\right)\right)\left(m_{U}\left(t-1\right)+\sum_{V}J_{UV}\frac{\partial m_{V}\left(t-1\right)}{\partial J_{UV}}\right). \end{array}$$

In Figure 2 we observe an example for a infinite-size kinetic Ising model with just one homogeneous region U with self couplings of value $J_{\mathrm{UU}}=J/N$. In this case, the model displays a critical point at J=1.

We argue that this critical phase transitions offers an interesting case for studying integrated information. First, systems at criticality display long-range correlations and maximal energy fluctuations and at the critical point [7,8], which should produce maximal dynamical integration, as noted in [16]. However, IIT 3.0 is concerned not with dynamics, but with causal interactions (i.e., how the states of mechanisms generate information by constrain future/past states), thus assuming critical dynamics is not enough for expecting maximum integration in the terms of IIT 3.0. Still, we can argue that (a) phase transitions in an Ising model mark a discontinuity between different modes of operation and (b) the critical point is characterised by maximum susceptibility (i.e., sensitivity to changes in intensive physical properties of mechanisms, e.g., Figure 2B) in front of external perturbation [7]. Because of this, when measuring integrated information in Ising models, we expect critical phase transitions to be observable in terms of integrated information, and critical points to have distinguishable properties respect other points of the phase space.

Using this toy model, we explore different aspects of the mathematical definitions of integrated information and the assumptions behind these definitions, using critical phase transitions as a reference. We will compute φ(τ) as follows. First, we select the initial state s(t). For finding a representative initial state, we start from a uniform distribution of P(s) (and the corresponding PI(S)=12NNS) and update until it stabilizes (using Equation (Equation 11) or Equation (Equation 12) for the finite and infinite cases, respectively). Then, we choose S(t)=argmaxP(S). From there, we update the probability distributions forward or backwards τ times with and without applying a partition and compute the distance between distributions for computing $\varphi_{\mathrm{effect}}$ and $\varphi_{\mathrm{cause}}$. Total integrated information φ will be computed as the minimum between the two. In all sections, we will assume that mechanism and purview contain the same units ${\left\{i\right\}}_{i\in I_{M_{t}}}={\left\{i\right\}}_{i\in I_{P_{t\pm \tau }}}$. We also will assume that the mechanism and the purview are composed by the whole system under analysis, except for Section 3.4 and Section 3.5, were smaller subsystems are analysed. Only in Section 3.5 do we compute the value of Φ, and we will argue that in our examples the first level of integration φ is enough for describing the behaviour of the system.

### 3.1. Dynamics and Temporal Span of Integrated Information

First, we explore integrated information of the effect repertoire of a system for a time span τ, $\varphi_{\mathrm{effect}}\left(\tau \right)$. In IIT 3.0, integrated information is defined as the level of irreducibility of the causal structure of a dynamical system over one time step [1]. The level of irreducibility is computed by applying partitions over the system, in which noise is injected in the connections affected by the partition. This is done by manipulating the transition probability matrix of the system. In previous work, integrated information has been applied to different temporal spans by changing the temporal graining of the system and joining consecutive states in a new Markov chain [12] or by concealing the micro levels in black box mappings [17]. Another possibility could be describing the transition probability matrix of the system from *t* and t+τ. However, is this the adequate way to capture integration at larger timescales? As IIT 3.0 operates with the transition probability matrix of a system, one could compute this matrix from time *t* to time t+τ and compute a new transition probability matrix for a bipartition by injecting noise in the connections affected by it at time *t*. This implies that noise is injected at the first step (at time *t*) and then the system behaves normally for the following steps. We will refer to this way of applying a partition as an ’initial noise injection’ (in contrast with a “continuous noise injection”, see below).

We explore this by computing integrated information with only one region of size N=256, with coupling values $J_{ij}=J/N$. If we compute φ for different values of τ (Figure 3A), we observe that for different couplings *J* integrated information always peaks at the ordered side of the phase transition. As τ is increased, this peak moves towards the critical point and its size decreases decreases, tending to zero. The assumption of an initial noise injection yields φ(τ)=0 at the critical point and maximum integration at the ordered side of the phase transition. Thus, integrated information in this case is not able to characterise the phase transition of the system

A different metric can be defined if, instead of applying the partition just at the initial step, we apply it to all τ updates (Figure 3B). We will refer to this way of applying a partition as a “continuous noise injection”, in contrast with the case in which noise is only injected at the first step. We propose that this is a more natural way to apply a partition, capturing larger integrated information around the critical point as we consider larger timescales. Moreover, as opposed to the previous case, which captured zero integration in the disordered side, this measure is able to capture increasing integration as the system approaches the critical transition from any side. One may note that in the mean-field approximation for infinite size (as shown in [14] and also Figure 5A), integration is zero when approaching a critical point from the disorder side. This is not a problem of the measure but a characteristic of the system, in which units have independent dynamics until *J* reaches the threshold of the critical point. For finite size, units are not completely independent and our measure correctly captures non-zero integration.

Still, some important considerations need to be taken into account when applying a continuous noise injection. In a initial noise injection, φ decreases with time as the effect of causal structures is diluted with time. In contrast, a continuous noise injection accumulates the effects of each time step, making integration grow for larger temporal span. These are very different assumptions, but we propose that the latter is more appropriate in our case in order to capture the long-range correlations and critical slowing down properties displayed by systems at criticality. Therefore, for the remainder of the article, we will assume a continuous noise injection.

### 3.2. Integrated Information of the Cause Repertoire

In the previous section we have explored the behaviour of $\varphi_{\mathrm{effect}}$ around a critical phase transition, i.e., the value of integrated information for the repertoire of states generated by the mechanisms of a system at time t+τ. IIT 3.0 proposes that integrated information should be computed as the minimum between $\varphi_{\mathrm{effect}}$ and $\varphi_{\mathrm{cause}}$ (Equation (Equation 4)). This is motivated by the “intrinsic information bottleneck principle”, proposing that information about causes of a state only exist to the extent it also can specify information about its effects, and vice versa [1].

Describing the cause repertoire is more complicated than the effect repertoire. IIT 3.0 [1] (Text S2) proposes to tackle the problem of defining P(s(t−1)|s(t)) by assuming a uniform prior distribution of past states $P_{U}\left(s\left(t-1\right)\right)$. This takes the form
(17)$$P\left(s\left(t-1\right)|s\left(t\right)\right)=\frac{P\left(s\left(t\right)|s\left(t-1\right)\right)P_{U}\left(s\left(t-1\right)\right)}{\sum_{s(t-1)}P\left(s\left(t\right)|s\left(t-1\right)\right)P_{U}\left(s\left(t-1\right)\right)},$$
where $P_{U}$ stands for a uniform probability distribution. This is equivalent to
(18)$$P\left(S\left(t-1\right)|S\left(t\right)\right)=\frac{P\left(S\left(t\right)|S\left(t-1\right)\right)P_{I}\left(S\left(t-1\right)\right)}{\sum_{S(t-1)}P\left(S\left(t\right)|S\left(t-1\right)\right)P_{I}\left(S\left(t-1\right)\right)},$$
where PI(S)=12NNS is the binomial distribution resulting from combining *N* independent distributions (obtained directly from $P_{U}\left(s\right)$). Similarly, $P_{\mathrm{cut}}\left(s\left(t-1\right)|s\left(t\right)\right)$ and $P_{\mathrm{cut}}\left(S\left(t-1\right)|S\left(t\right)\right)$ can be computed like in Equations (Equation 17) and (Equation 18), assuming a modified conditional probability $P_{\mathrm{cut}}\left(S\left(t\right)|S\left(t-1\right)\right)$

What is the effect of considering an independent prior? As we observe in Figure 4A, for τ=1, integration remains high even for large values of *J*. As we increase τ, integration decreases. This behaviour is completely different to the effect repertoire (Figure 3B). Intuitively, such a difference of behaviour from cause and effect mechanisms is strange for an homogeneous system in a stationary state as the one under study here. More importantly, the measure of $\varphi_{\mathrm{cause}}$ fails to capture integration around the critical point, and displays the largest values of integration of the system far into the ordered side of the phase space. Note that as $\varphi =\min(\varphi_{\mathrm{cause}},\varphi_{\mathrm{effect}})$, in this case, the value of integration would be dominated by the cause repertoire, and φ would not diverge around the critical point.

It is possible to drop the assumption of an independent prior, but some assumption about the prior distribution is needed. A simple alternative is to assume that the system is in a stationary state with distribution $P_{\mathrm{st}}\left(s\left(t\right)\right)=P_{\mathrm{st}}\left(s\left(t-1\right)\right)$, having then
(19)$$\begin{array}{cc} P(s(t-1)|s(t)) & =\frac{P\left(s\left(t\right)|s\left(t-1\right)\right)P_{\mathrm{st}}\left(s\left(t-1\right)\right)}{\sum_{s(t-1)}P\left(s\left(t\right)|s\left(t-1\right)\right)P_{\mathrm{st}}\left(s\left(t-1\right)\right)}=\frac{P\left(s\left(t\right)|s\left(t-1\right)\right)P_{\mathrm{st}}\left(s\left(t-1\right)\right)}{P_{\mathrm{st}}\left(s\left(t\right)\right)}, \end{array}$$
(20)$$\begin{array}{cc} P(S(t-1)|S(t)) & =\frac{P\left(S\left(t\right)|S\left(t-1\right)\right)P_{\mathrm{st}}\left(S\left(t-1\right)\right)}{\sum_{S(t-1)}P\left(S\left(t\right)|S\left(t-1\right)\right)P_{\mathrm{st}}\left(S\left(t-1\right)\right)}=\frac{P\left(S\left(t\right)|S\left(t-1\right)\right)P_{\mathrm{st}}\left(S\left(t-1\right)\right)}{P_{\mathrm{st}}\left(S\left(t\right)\right)}, \end{array}$$


In this case, computing $\varphi_{\mathrm{cause}}\left(\tau \right)$ (Figure 4B), we observe that the integration of the cause and effect repertoires has a similar behaviour as *J* changes, yielding similar curves to $\varphi_{\mathrm{effect}}\left(\tau \right)$. Still, note that integration values are slightly lower for the cause repertoire.

Thus, the assumption of an independent prior has dramatic consequences, which can be avoided by assuming an stationary distribution. Another alternative for systems undergoing a transient is to compute the trajectory of probability distributions P(s(t)) and use it as priors, though this makes the computation much more costly. For the rest of the manuscript, we will assume an stationary prior. Note that the noise injected when partitioning the system is still uniform in all case.

### 3.3. Divergence of Integrated Information: Wasserstein and Kullback–Leibler Distance Measures

We have seen that, using our assumptions, φ grows with τ around the critical point in a finite system, suggesting that the value of integration may diverge in the thermodynamic limit. We test this divergence by computing integrated information φ for for networks of different size *N* and a given τ. In general, the relationship of φ and τ is complex, as for each value of *J*, it depends on the transient dynamics of the system. It is not the goal of this article to explore this issue in detail, but we want to ensure that finite systems have enough time to get close to a stationary regime. Thus, from now on, for simplicity, we will use a value of $\tau =10\log_{2}N$, where *N* is the size of the system. We choose this relation because we have tested that it ensures the divergence of integrated information around critical points, although other relations we tested (e.g., τ∝N) maintain the qualitative results shown in the following sections.

To test the divergence of φ, we compute the value of integrated information for the largest mechanism of a kinetic Ising model with an homogeneous region with different size *N*, and assuming continuous noise injection and a stationary prior. We observe in Figure 5A that for finite sizes $\varphi_{\mathrm{cause}}$ (black line) shows a diverging tendency around the critical point. Effect integration $\varphi_{\mathrm{effect}}$ (grey line) shows a similar divergence, with values slightly larger. In this case, we also computed the value of $\varphi_{\mathrm{effect}}$ for infinite size. When N→∞, units $s_{i}\left(t+\tau \right)$ become independent, and the Wasserstein distance of a system with one region is computed analytically as $D_{W}\left(P\left(s\left(t+\tau \right)|s\left(t+\tau \right)\right)|\right|P_{\mathrm{cut}}\left(s\left(t+\tau \right)|s\left(t+\tau \right)\right))=\frac{1}{2}\frac{\partial m(t+\tau )}{\partial J}J$ (see Section B.2). The divergence of $\varphi_{\mathrm{effect}}$ for infinite size shown in Figure 5A was also analytically characterised in [14]. As $\varphi_{\mathrm{effect}}$ is always larger than $\varphi_{\mathrm{cause}}$, in this case the total integration is always $\varphi =\varphi_{\mathrm{cause}}$. Summarising, we can conclude that φ computed using the Wassertein distance shows a divergence around the critical point of the kinetic Ising model.

Many versions of φ use the Kullback–Leibler divergence as an alternative to the Wasserstein distance. As seen in Figure 5B, this change can lead to an important difference in the results of $\varphi_{\mathrm{cause}}$ (black line) and $\varphi_{\mathrm{effect}}$ (grey line). The figure shows that φ tends to peak around the critical point but decreases with the size of the system. Also for this case, $\varphi =\varphi_{\mathrm{cause}}$ for the cases we computed. By doing a similar approximation than in the previous case can be used to compute $\varphi_{\mathrm{effect}}$ in the infinite case (see Appendix C), using the well-known relation between the Kullback–Leibler and Fisher information, it can be shown that $D_{KL}\left(P\left(s\left(t+\tau \right)|s\left(t+\tau \right)\right)|\right|P_{\mathrm{cut}}\left(s\left(t+\tau \right)|s\left(t+\tau \right)\right))=\frac{1}{2}\frac{1}{1-m^{2}\left(t+\tau \right)}J^{2}\frac{1}{N}{\left(\frac{\partial m(t+\tau )}{\partial J}\right)}^{2}$, tending to zero for diverging size *N* (see Appendix C). Using this expression, we find that for infinite size the value of $\varphi_{\mathrm{effect}}N$ diverges. However, computing the values for the finite networks, we find that $\varphi_{\mathrm{effect}}N$ and $\varphi_{\mathrm{cause}}N$ do not diverge for finite values of *N* (Figure 5C). This can be interpreted as a similar phenomenon found in homogeneous Ising models (e.g., Curie–Weiss model [13]), where the Heat Capacity, equivalent to the Fisher approximation to the Kullback–Leibler divergence computed here, does not diverge for finite sizes as the size of the system grows.

These results illustrate that different distance measures can have important effects in the behaviour of φ. As well, our results show that different metrics can hold different relations between finite models and the mean-field behaviour of the model with infinite size. For the Wasserstein distance, φ in finite systems tends to a diverging behaviour around the critical point, characterised for the infinite mean-field model. Conversely, for the Kullback–Leibler divergence, φ does not diverge in finite models and it does diverge for the mean-field infinite model (for the effect repertoire). In this case, the symmetry breaking of the system for infinite size provokes that the behaviour of the system is different in the mean-field model (a similar phenomena takes place with simple measures as the average magnetisation). This effect can be relevant for studying φ in real finite systems by computing their mean-field approximations.

Although most versions of integrated information measures used the Kullback–Leibler divergence, recently IIT 3.0 suggested that the Wasserstein distance is a more appropriate measure [1] (Text S2). The result presented here show that the change of distance measure can have important implications when measuring large systems. Further work should inspect how different distance measures are able or not to capture the scaling behaviour of different systems and how is this coherent with the properties of the systems under study. A way to do so could be to explore the connection between the relation of the Wasserstein and Kullback–Leibler versions of integrated information with well-known variables in Ising models like the magnetic susceptibility or the heat capacity of the system (see Appendix B and Appendix C).

As we have shown that under the appropriate assumptions both $\varphi_{\mathrm{cause}}$ and $\varphi_{\mathrm{effect}}$ have similar diverging tendencies, in the rest of the example, we will not show these variables separately and will just show $\varphi =\min(\varphi_{\mathrm{cause}},\varphi_{\mathrm{effect}})$.

### 3.4. Situatedness: Effect of the Environment of a System

In IIT 3.0, there is a difference between a *mechanism*, where a first level of integration φ is computed, and a *candidate set*, composed of different mechanisms, where a second-order level of integration Φ applies. When computing integration at these two levels, there are some assumptions about how the elements outside of the systems are considered. In IIT 3.0, the elements inside the candidate set but outside of the mechanism are treated as independent sources of noise. Respectively, elements outside the candidate set are treated as background conditions, and are considered as fixed external constraints.

What are the effects of these assumptions when computing integrated information in a critical phase transition? We measure again integration of a kinetic Ising model with one region of size *N* and coupling *J*. However, instead of considering the whole system, we measure the level of integration of a subsystem or mechanism M covering a fraction of the system M/N, where *M* is the size of the mechanism. We choose a value of $M=\frac{3N}{4}$, although other fractions yield similar results. To compute Equation (Equation 11) with and without the partition, we divide the system in two regions: one consisting on the units belonging to the mechanisms, and the other containing the units outside the mechanism. We measure the integrated information of the mechanism $\varphi_{M}$ under three different assumptions: (a) that units outside of the mechanism operate normally, (b) units outside the mechanism are independent noise sources and (c) units outside the mechanism are fixed as external constraints.

In the first case, when external units operate normally (Figure 6A), we observe that the divergence of $\varphi_{M}$ is maintained (although testing different values of *M* shows that $\varphi_{M}$ increases with the size of the mechanism, see [14]). In contrast, if we accept the assumptions of IIT 3.0 and take the elements outside the mechanism as independent sources of noise or as static variables, the behaviour of $\varphi_{M}$ changes radically. In the former case, when outside elements are independent noise sources the divergence is maintained but takes place at a different value of the parameter *J* (Figure 6B). This happens because inputs from uniform independent sources of noise will be distributed around a zero mean field value, and thus the phase transition of the system takes place at larger values of *J* that compensate for the units that are now uncorrelated. Thus, considering the elements outside of the mechanism as independent sources of noise can be misleading, showing that maximum integration takes places at different points of the system. In this case, the position of the divergence is located at larger values of *J*, corresponding with significantly lower values of covariance and fluctuations in the units of the system, therefore not reflecting the actual operation of the mechanisms.

The latter assumption implies maintaining the state of the units outside of the mechanism with the static values that they had at time *t*. In this case (Figure 6C), we find that $\varphi_{M}$ does not diverge, and instead it has a peak in the ordered side of the phase transition. We can understand this by thinking that the effect of constant fields is equal to adding a value of $H_{i}$ equal to the input from static units, therefore breaking the symmetry of the system and precluding a critical phase transition. In both cases, we observe that ignoring or simplifying the coupling between a system and its environment can affect drastically the values of integration as a system scales.

### 3.5. System-Level or Mechanism-Level Integration: Big Phi versus Small Phi

So far, we analysed the behaviour of integration measures φ describing the integration of mechanisms of a system. IIT 3.0 postulates that the integration of a system is defined by a second-order measure of integration, Φ, applied over the set of all its mechanisms. To explore how Φ behaves for systems of different sizes, we compute it for a homogeneous system with one region, including the different modifications assumed in the previous subsections. Note that this modifies the measure, but it still allows us to inspect some of its scaling aspects.

For measuring Φ, first φ is computed for the different mechanisms of the system, and then the integration of the set of mechanisms is compared with the set of values of φ of the system under unidirectional partitions, using a modified Wasserstein distance [1] (Text S2). In the case of a homogeneous system with just one region, the MIP is any of the partitions that isolates one single node from the rest of the system. The value of Φ is the modified Wasserstein distance (with and without applying the MIP) between the values of φ of the set of mechanisms of the system.

In Figure 7, we compare the values of φ of the larger mechanism (Figure 7A) with the normalised values of Φ of the whole system (Figure 7B), for a homogeneous system with only one region with self-couplings *J*. The value of φ of the larger mechanisms diverges around the critical point as expected. In the case of Φ, we find that for all values of *J* integration grows with size very rapidly. This is due to the fact that the number of concepts (i.e the number of mechanisms) of the system grows exponentially with size. The number of mechanisms or concepts is NC=∑k=1NNk=2N−1. Thus, we normalise the value of Φ dividing by $N_{C}$ (Figure 7B). Using normalised values of Φ, we observe that the system still diverges at the critical point. Furthermore, in this case the divergence is faster than in the case of φ, as it accumulates the effects of the divergence of many mechanisms under a second partition.

If we observe the contribution of different mechanisms to Φ, we observe that most of the contributions to Φ are determined by the mechanisms affected by the MIP. In this case, all the value of φ is transported by the Wasserstein distance into a new point defined by an independent distribution [1] (See Text S2) (e.g., for N=128 around 98% of the value of Φ is defined by the value of φ of the mechanisms under the MIP).

In our example, it seems that the relation between φ and Φ is not quite relevant (the divergence of the later seems to be an amplified version of the former). Heterogeneous or sparsely connected systems may present more complicated relations and present important differences in the behaviour of the highest order φ and the total Φ. Still, we believe that our simple example calls for a better justification of the need of measuring a second-order level of integration in IIT 3.0 and the difference of the two levels respect well-studied properties of systems.

### 3.6. Values versus Tendencies of Integration

Finally, we explore mechanism integration φ in the case of two homogeneous regions: one region A with self-interaction and another region E, which is just coupled to the first without recurrent connections (i.e., $J_{\mathrm{EE}}=0$, Figure 8A). This case was used in [14] to represent an agent interacting with an environment, exploring the power of integrated information to delimit what is the most integrated part of the system. This delimitation has been proposed to identify the autonomy of small biological circuits [2], but it is still unclear if the conclusions of analysis in such small systems and simplified models could be extended to larger models of neural and biological systems.

For different values of recurrent connections $J_{\mathrm{AA}}=J_{R}$, two values of bidirectional couplings $J_{\mathrm{AE}}=J_{C}$, $J_{\mathrm{EA}}=2J_{C}$ are tested: $J_{C}=0.8$ and $J_{C}=1.2$. The results in [14] showed that, for infinite sizes, in the weaker coupling condition $J_{C}=0.8$, A was the most integrated unit of the system at the critical point. In contrast, for a stronger coupling $J_{C}=1.2$, the joint AE system was the one that presented higher integration for infinite size. In Figure 8, we show the values of integration of A and AE for different sizes (integration of E is always zero as there are no recurrent connections for this region), with $J_{C}=0.8$ (Figure 8B,C) and $J_{C}=1.2$ (Figure 8D,E).

For $J_{C}=0.8$, we observe that $\varphi_{A}$ is always larger, independently of the size of the system, showing that A is always more integrated than AE. However, for $J_{C}=1.2$, we find an interesting behaviour. We can observe that for small sizes (N=8,16) A is more integrated. Conversely, for larger sizes (N=64,128) we observe that AE is more integrated, as its value of $\varphi_{A}$ diverges faster with size than $\varphi_{\mathrm{AE}}$.

This is relevant because in many cases integrated information can only be measured for rather small systems. When analysing models of real neural or biological systems, these should be coarse grained or discretised in order for φ measures to be applicable. In such cases, we can expect that the delimitation of the most integrated units of the system have different values than at larger scales. Thus, rather than the exact value of φ, the diverging tendencies in the model might be most informative about the behaviour of the real observed system when small networks are considered.

## 4. Discussion

In this article, we critically reviewed different aspects of the definition of integrated information proposed by IIT 3.0, exemplifying them in toy models displaying critical phase transitions. Using a homogeneous Ising model, we simplify the calculations to measure integrated information in large systems. It is well known from theory in spin glasses that the infinite range homogeneous Ising model (also known as Curie Weiss model) presents a critical point for J=1 and H=0 [13]. Although we argue that critical phase transitions should be observed in integrated information measures (as critical points display long-range correlations, maximal susceptibility to parametric changes and preserve integrative and segregative tendencies of the system, see [7,16]), we have shown how different aspects of the definition of φ prevent to capture the critical phase transition of the system as it grows larger in size. This investigation has led us to propose reformulations of some aspects of the theory in order to address some of the problems encountered during the study.

As IIT 3.0 has been mostly tested in small logic gate circuits, exploring the behaviour of integrated information in large Ising models has allowed us to investigate questions that were so far unexplored and inspect some of the assumptions of the theory from a new perspective. We consider that the value of the study is twofold. On one hand, we propose a family of models with known statistical properties, where calculations of integrated information are simplified. These and similar models could work as a benchmark for testing properties of integrated information in large systems. On the other hand, the reformulations of different aspects of the theory proposed during the paper could be considered by future versions of IIT, to capture some of the phenomena that we could expect in large, complex systems.

First, we explored how the application of integrated information over an adequate timescale is important to observe increasing values of integration as the system scales. The dynamics of the Ising model are characterised by a “critical slowing down” as the critical point is approached. Consequently, we observed that to capture critical diverging tendencies, timescales larger than one time step should be used. As the dynamics of critical system display correlations at very different timescales, and the span of these timescales increase with the size of the system, integrated information should be evaluated in a way that the diversity of timescales is captured. In our analysis, we found that the way to capture integration near critical points is to apply partitions in a different way than IIT 3.0. In IIT 3.0 partitions are applied by injecting noise in the input state of the system and then computing the forward and backwards distributions, but this approach did not capture the phase transition in the model. In contrast, we successfully characterised the phase transition as diverging integrated information around the critical point by applying several updates of the state of the system and injecting noise at each update.

Second, to capture the cause repertoire of a state (integration in the causal dependencies of a mechanism with previous states), IIT 3.0 proposes to assume a uniform prior distribution of past states. We show that this assumption can distort the observed values of integration, losing an adequate characterisation of the critical phase transition. We suggest that the real prior distribution (either stationary or transient) should be used if cause repertoires are considered.

The third aspect we studied is the use of different distance measures between probability distributions. Specifically, we compared the Wasserstein distance used by IIT 3.0 with the Kullback–Leibler divergence, which is the choice for many competing definitions of integrated information. First, we show that values of the Kullback–Leibler divergence should be weighted by the size of the system in order to be comparable to the Wasserstein distance under the MIP; otherwise, they tend to zero as the system grows. We also show that, in a homogeneous kinetic Ising model at criticality, the Wasserstein distance shows diverging tendencies for finite sizes, whereas the Kullback–Leibler divergence only shows a finite peak. This shows that, in some cases, the Wasserstein distance may detect some divergences that would be ignored by the Kullback–Leibler divergence. Still, it should be debated whether it is adequate that a system like the toy model presented here shows a diverging value of integration. A closer examination of the behaviour of known quantities in an Ising model could constitute an adequate starting point for this discussion. In this sense, the results of the Wasserstein distance and Kullback–Leibler divergence can be connected with the behaviour of known quantities in the homogeneous Ising model. For example, the susceptibility of the system diverges at the critical point while the heat capacity of the system only shows a peak [13]. Both measures can be related to approximations of φ using the Wasserstein and Kullback–Leibler measures, respectively (from Equations (Equation 42) and (Equation 29)).

Furthermore, we analysed a crucial aspect of integration measures that is often overlooked: the situatedness of the system. The central claim of situated approaches to cognitive behaviour is that the agent-environment coupling shapes brain dynamics in a manner that is essential to behavioural or cognitive functionality [18,19]. Thus, ignoring or dismissing this brain-body-environment coupling can result in a substantial quantitative and qualitative distortion of the activity of a neural system [20]. Besides, there are deep theoretical reasons that come from the enactive perspective on cognition that establish that the very notion of cognitive performance is world-involving, i.e., that it is co-constituted by agent and environment [21]. In contrast, IIT 3.0 dismisses the bidirectional interaction between the system under evaluation and its environment for computing integration, with the aim to assess the integrated information from the “intrinsic perspective” of the system itself. Specifically, IIT 3.0 considers the units outside the system (i.e., outside the candidate set) as static variables and the units within the system but outside the evaluated mechanism as independent sources of noise. We show in the model that both assumption can have dramatic effects in the behaviour of the system. The assumption of static variables makes the divergence at the critical point disappear, and the assumption of independent sources of noise creates spurious divergences of integrated information at different positions than the original model. Only a situated version of integrated information, which does not dismiss the activity of the environment and its couplings to the system, can correctly measure integrated information even for a model as simple as ours. This suggest that the intrinsic notion of information or the intrinsic perspective of a system cannot dismiss the system’s regulation of its coupling with the environment [22]. Thus, ignoring the coupling with the outside of a sytem can have important consequences for the application of integrated information measures in simulated and experimental setups. For example, in [23], different agents are characterised by the integrated information of its neural mechanisms, but ignoring the environment might miss important channels of sensorimotor coordination contributing to the integration of the system. Similarly, attempts to identify the physical substrate of consciousness in brains [24] should take into account situated and embodied aspects of brain activity, or even consider the possibility that (at least at some moments) this substrate can cut across the brain–body–world divisions, rather than being confined inside the brain [25].

In other experiments, we compared the differences between the values of mechanism-level and system-level integration (φ and Φ) in a homogeneous system with one region, finding that some normalisation constants are required to compare Φ of systems with different size. We also found that Φ also diverges at the critical point, and it does faster than φ, due to the second partition applied and the accumulation of the different mechanisms of the system. Although here we compute Φ for a very simple system, we suggest that the introduction of this second level of analysis should be better justified. In that sense, recent work explores very small networks showing how the compositional definition of measures like Φ can yield very different results than the non-compositional mechanism-level measures φ [26]. Further work could try to better characterise the difference between the two levels in systems with analytically tractable properties like the Ising systems with multiple regions presented here.

Finally, we compared the diverging tendencies of two coupled subregions, showing that the delimitation of integrated information might change with size as the integration of some regions diverges faster than others. This is specially relevant as IIT 3.0 gives a prominent relevance to the areas of the brain with maximal integration (the “neural substrate of consciousness”). If integration takes the form of diverging values of φ around certain classes of critical points, or regions (see [14]), then the neural substrate supporting maximal integration should be characterised by how fast integration diverges with size, and not by the value of integration yielded by simplified models (e.g., by coarse-graining observed time series), which can be potentially misleading.

These results exploring homogeneous kinetic Ising models show that the calculation of integrated information presents important challenges even in simple models. This work serves to demonstrate that the measure is very susceptible to design assumptions and that the behaviour of the measure changes drastically because of this. In this scenario, we show how the connection between the theory (IIT 3.0), a theoretical understanding of complex dynamical systems (critical phase transitions), and the study of simplified models exemplifying known phenomena (homogeneous Ising models) offers a path to systematically study the implications of these assumptions. Our results compel researchers interested in IIT and related indices of complexity to apply such measures under careful examination of their design assumptions. Rather than applying the measure off-the-shelf, ideally researchers should be aware of the assumptions behind the measure and how it applies to each situation. In this way, theory can go in hand with cautious experimental applications, avoiding potentially misleading interpretations and ensuring that they are used to improve our understanding of biological and neural phenomena.

## Figures and Tables

**Figure 1 entropy-21-01198-f001:**
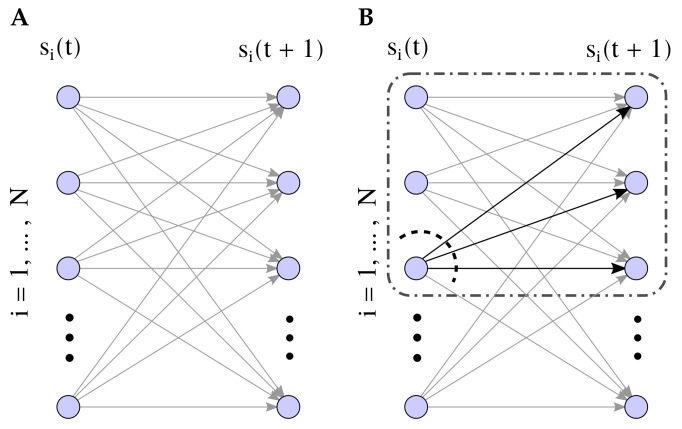
(**A**) Description of the infinite size kinetic Ising model. (**B**) Description of the partition schema used to define perturbations. Partitioned connections (black arrows) are injected with random noise. Nonpartitioned connections operate normally or are independent sources of noise (see Section 3.4).

**Figure 2 entropy-21-01198-f002:**
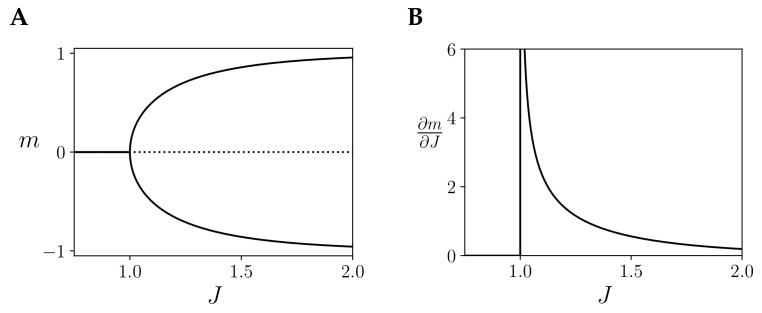
Description of the behaviour of the homogeneous Ising model with one region and coupling *J*, showing a critical point at J=1. (**A**) Values of mean firing rate *m* for the stationary solution of the kinetic Ising model with one homogeneous region. (**B**) Value of $\frac{\partial m}{\partial J}$ for the positive stationary solution of the kinetic Ising model with one homogeneous region, diverging at the critical point.

**Figure 3 entropy-21-01198-f003:**
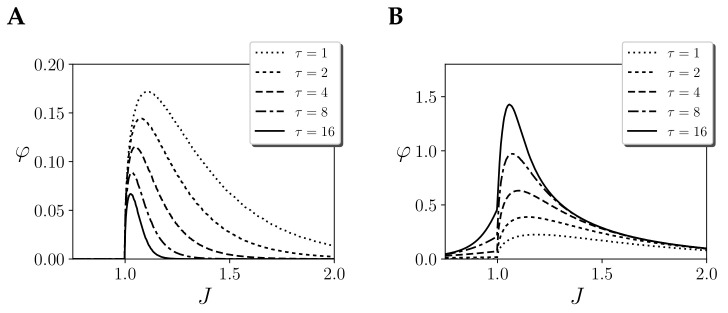
Integration of the effect repertoire $\varphi_{\mathrm{effect}}\left(\tau \right)$ of the largest mechanism of a homogeneous Ising model with one region of size N=256 and couplings *J* with different temporal spans τ, assuming (**A**) initial injection of noise and (**B**) continuous injection of noise. Note that τ=1, in both cases $\varphi_{\mathrm{effect}}$, has the same value.

**Figure 4 entropy-21-01198-f004:**
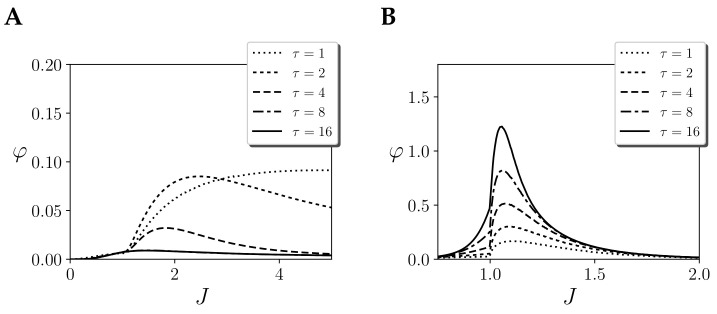
Integration of the cause repertoire $\varphi_{\mathrm{cause}}\left(\tau \right)$ of the largest mechanism of a homogeneous Ising model with one region of size N=256 and couplings *J* with different temporal spans τ, assuming (**A**) an independent prior and (**B**) the stationary distribution as a prior. Continuous noise injection is assumed.

**Figure 5 entropy-21-01198-f005:**
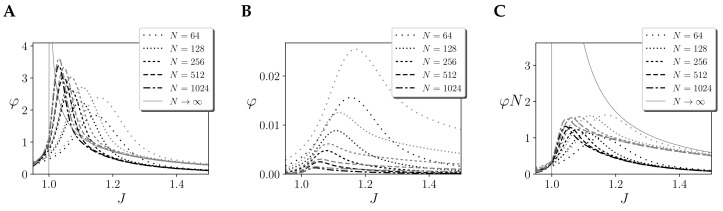
Integrated information φ(τ) for the cause (black lines) and effect (grey lines) repertoires of the largest mechanism of a homogeneous kinetic Ising models with one region of size *N* (and infinite size when N→∞) and coupling *J* using (**A**) the Wasserstein distance. (**B**) The Kullback–Leibler divergence, and (**C**) values of φN using the Kullback–Leibler divergence. Note that in all cases $\varphi \left(\tau \right)=\varphi_{\mathrm{cause}}\left(\tau \right)$. All cases are computed with $\tau =10\log_{2}N$ for finite systems and τ→∞ for infinite systems. Continuous noise injection and a stationary prior are assumed.

**Figure 6 entropy-21-01198-f006:**
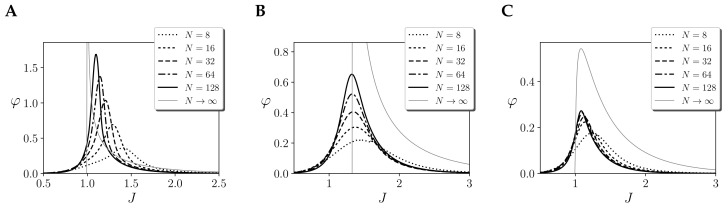
Effects of the environment in integrated information. Integrated information $\varphi_{M}\left(\tau \right)$ (black lines) of a mechanism M of size $\frac{3N}{4}$ of a homogeneous kinetic Ising model with one region of size *N* and coupling *J*, assuming that elements outside of the mechanism operate (**A**) normally, (**B**) as independent sources of noise and (**C**) as static input fields. Values of $\varphi_{M}\left(\tau \right)$ are compared with $\varphi_{M,\mathrm{effect}}\left(\tau \to \infty \right)$ (grey line) to show diverging tendencies of the effect repertoire. Note that tendencies of $\varphi_{M,\mathrm{effect}}\left(\tau \to \infty \right)$ are larger than values of $\varphi_{M}\left(\tau \right)$, as the effect repertoire tends to show larger values. Values of φ are computed with $\tau =10\log_{2}N$ for finite systems and τ→∞ for infinite systems. Continuous noise injection and a stationary prior are assumed.

**Figure 7 entropy-21-01198-f007:**
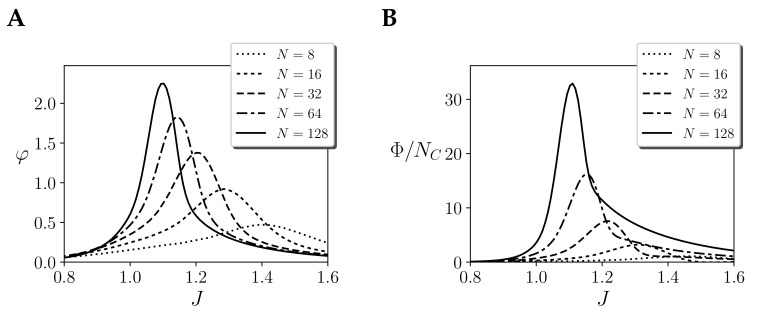
Mechanims and system-level integration in a homogeneous system with one region of size *N* and coupling *J*. Values of (**A**) φ of the largest mechanism and (**B**) values of Φ for the whole system. Measures with $\tau =10\log_{2}N$, assuming continuous noise injection, stationary priors and environment coupling.

**Figure 8 entropy-21-01198-f008:**
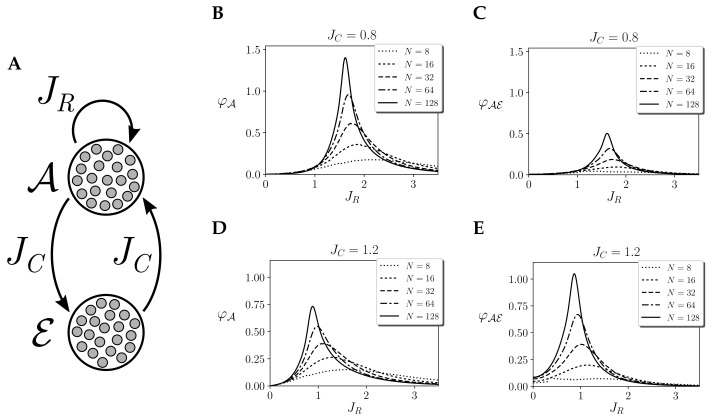
Integrated information in a system coupled to an environment. (**A**) Structure of couplings between the two regions A,B of size $N_{A}=N_{E}=\frac{N}{2}$ of a homogeneous kinetic Ising models with couplings $J_{\mathrm{AA}}=J_{R},J_{\mathrm{EE}}=0,J_{\mathrm{AE}}=J_{C},J_{\mathrm{EA}}=2J_{C}$. (**B**–**E**) Integrated information of the mechanism A, $\varphi_{A}$ and mechanism AE, $\varphi_{\mathrm{AE}}$, for values of $J_{R}=1$ and $J_{C}=0.8$ and $J_{C}=1.2$, respectively. Values of φ are computed for $\tau =10\log_{2}N$ for finite systems and τ→∞ for infinite systems. Continuous noise injection, stationary priors and environment coupling are assumed.

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
