# Peer review of "Scaling Behaviour and Critical Phase Transitions in Integrated Information Theory"

_entropy, 2019, doi:10.3390/e21121198_

Round 1

Reviewer 1 Report

Integrated Information Theory is aimed to define a measure of conscious activity (Φ), characterized as the irreducibility of a dynamical system to the sum of its components.

Grammar – “is aimed to define” is awkward.

Due to its computational cost, current versions of the theory (IIT 3.0) are difficult to apply to systems larger than a dozen of units, and in general it is not well known how integrated information scales in time and space.

This makes it sound as if IIT has never had any work on how integrated information changes across space or time – whereas Hoel et al. 2016 “Can the macro beat the micro? Integrated information across spatiotemporal scales” shows that integrated information can increase at higher scales. I think the author means how the calculation of integrated information changes as systems get larger in size.

As a result, we point to some aspects of the mathematical definitions in IIT 3.0 that are flawed to capture critical phenomena and propose a reformulation of some aspects of the theory.
Grammar – it doesn’t make sense to point to flawed definitions to capture critical phenomena.

Integrated Information Theory (IIT, [1]) was developed to address the problem consciousness by providing a measure of integration, Φ, that quantifies to what extent a dynamical system generates information that is irreducible to the sum of its parts considered independently.
Grammar – the problem of

(displaying a diversity locally specific patterns)

Grammar

Furthermore, IIT attempts to quantify the irreducibility of these complex patterns, revealing the boundaries in the organization (i.e. delimiting the parts of the system that are integrated into a functional unit) of complex dynamical systems

IIT doesn’t quantify the complexity of patterns, it quantifies the causal structure. These are very different, for instance, see Larissa et al. 2016 “The intrinsic cause-effect power of discrete dynamical systems.” Additionally, it’s unclear how IIT attempts to quantify the integrated information for complex dynamical systems, given that previously the author stated that IIT can only be run on small systems.

In IIT 3.0 [1], the lastest version of the framework proposed by the lab of Giulio Tononi

Spelling mistake. Additionally, I’m not sure this is the best way to reference the paper.

can only be computed for binary networks composed of up to a dozen of units. A variety of other measures exist to compute integrated information [3] and some of them are computationally lighter, but all share these limits to some extent.

What about all the metrics and adaptations that are used for brain imaging? Are those for binary networks as well? In what sense are large systems still a problem for the existing heuristics? There is not enough background to motivate these issues here.

Since most versions of Φ require computing distance measures between probability distributions of a system and finding the minimum information partition (MIP), they present important restrictions in terms of computational cost as a system scales in size.

Is the restriction really because of differences between probability distributions? Isn’t the MIP by far the limiting factor, given it is a bell number?

As a consequence, IIT measures have been limited to very small systems in general, and it is not well understood how they scale with the size or temporal scale of the system

Grammar – they

Previous work [4] has tried to analyse how integrated information changes with spatial and temporal coarse graining in small networks

Did they try and fail? Grammar.

 Aside from parctical difficulties due to its computational costs

Grammar

the measure of Φ might not be well defined

Grammar

In general, measures of Φ can only be applied to small networks

What about all the ones for brain imaging. They are just ignored consistently throughout the motivating section of the text.

dynamics its complex even for simple models

grammar

In the critical state, even in simple systems, dynamics are dominated by small bursts of local (i.e. segregated) activity, yet large avalanches of globally coordinated (i.e. integrated) activity

Grammar

Despite its appealing properties, critical phenomena are theoretically characterized for systems of infinite size, and can be characterized in large finite systems as divergent tendencies as they scale

Grammar

In the framework of IIT 3.0, it is required that a system satisfies the Markov property (i.e., the state at time t only depends on the state at time t − τ)

In Hoel et al. 2016 “Can the macro beat the micro? Integrated information across spatiotemporal scales” there are clearly systems that are second-order

and how it constraints the potential future or past states states.

Multiple grammatical mistakes

as the distance between the cause-effect structure of the system, and cause-effect structure defined by its minimum information partition

Grammar

we define a general model defining causal temporal interactions between variables

What would a causal non-temporal interaction be?

We divide system is divided into 125 different regions

Grammar

the MIP in most cases one of the partitions that cuts the lowest number of connections for each region.

Grammar

in general the MIP still is one that isolates one unit of one of the regions comprised by the mechanism.

Where is this shown “in general”

the level of irreducibility of the causal distributions

What are causal distributions?

As IIT 3.0 operates with the transition probability matrix of a system, one could compute this matrix from time t to time t + τ and compute a new probability transition matrix for a bipartition by injecting noise in the connections affected by it at time t. This implies that the system behaves normally for the following steps.

It’s unclear what’s going on here. More time needs to be explicitly spent saying why this is different, perhaps with an actual example figure if this difference is a key point of the paper.

The figure text from this figure is just lifted from the figure text of the previous figure. And there’s no figured 3 even cited in the text. And B in this “Figure 3” looks a lot the C in Figure 2, but cropped differently.

This behaviour is completely different to the one from in the effect repertoire. This should not happen for a homogeneous system in a stationary state as the one under study here.

Grammar. Also, how is the reader to know this isn’t due to your modifactions in the calculation of integrated information to adapt to Ising models rather than if one actually calculate using the real values from IIT 3.0, even though it is computationally intractable?

Also, again the measure of φcause fails to capture integration around the critical point.

What’s the argument behind this being necessary?

in [5, see Appendix B] We can see in Figure 4.A it is shown how the value of φ diverges with size.

Grammar

These results illustrate that different distance metrics can have important effects in the behaviour of φ.

This is an interesting result and should be highlighted a bit more.

We measure the integrated information of the mechanism φM under three different assumptions: a) that units outside of the mechanism operate normally, b) units outside the mechanism are independent noise sources, and c) units outside the mechanism are fixed as external constraints.

Again, this is quite interesting. Both the notion of exploring the difference between distributions and how to treat outside sources (as noise or so on) is extremely interesting. However, this is only explored in relationship to the Ising model. So the claims are somewhat incomplete, given the adaptations that are made. What about doing this is Markov chains and asking the same questions?

Second, some of the assumptions made by IIT 3.0 for computing integration over the cause repertoire of a state (the distribution of previous states) present problems for for capturing the integration at a critical point.

Grammar.

Plus, in the stationary case, the cause repertoire is identical to the effect repertoire, which allows to ignore the distinction.

Grammar.

GENERAL THOUGHTS 
The paper begins by arguing that IIT 3.0 doesn't scale in terms of its calculation and then introduces all sorts of assumptions in the Ising model to make it calculable. This doesn't actually solve the scaling problem so much as show that as long as enough assumptions are made, some version of IIT can be calculated. The paper begins by motivating itself by talking about scale but the takeaway is actually that assumptions in the calculation of IIT, specifically in terms of the chosen difference method and then also the choice of what distribution to use for elements outside the system, can be supported or disproven by watching how the measure operates in the Ising model. These are two mostly different problems, and yet the paper is motivated by the one that's not solved or really even argued for in the paper itself.

Overall, my impression of this paper is that it is a draft. There are numerous mistakes throughout involving grammatical errors, misplaced figures and figure texts. Two different times the figure texts of two different figures are identical. There is a figure included that is never cited. This is not a polished or final paper and should not have been submitted as such. There is a lot of repetition, and many sections could be a lot shorter. That said, some of the results are quite interesting and the general idea of the paper is based on an interesting argument – that PHI needs to be computed in a contextual manner dependent on the system. I do not believe it should be published in its current condition, but is due for a major revision that involves polishing and editing. In general the paper needs a stronger motivation, with the conclusions driving the original motivation.

Author Response

- We thank the reviewer for the thorough revision of the manuscript. We have corrected the grammar errors pointed here and others, and we have sent the manuscript for English proofreading for further revision (we will incorporate the corrections when they arrive).

- As well we have corrected other aspects of the manuscript. We comment the more relevant changes blow.

Due to its computational cost, current versions of the theory (IIT 3.0) are difficult to apply to systems larger than a dozen of units, and in general it is not well known how integrated information scales in time and space.

This makes it sound as if IIT has never had any work on how integrated information changes across space or time – whereas Hoel et al. 2016 “Can the macro beat the micro? Integrated information across spatiotemporal scales” shows that integrated information can increase at higher scales. I think the author means how the calculation of integrated information changes as systems get larger in size.

- We rephrased this in different places to refer to systems getting larger in size.

Furthermore, IIT attempts to quantify the irreducibility of these complex patterns, revealing the boundaries in the organization (i.e. delimiting the parts of the system that are integrated into a functional unit) of complex dynamical systems

IIT doesn’t quantify the complexity of patterns, it quantifies the causal structure. These are very different, for instance, see Larissa et al. 2016 “The intrinsic cause-effect power of discrete dynamical systems.” Additionally, it’s unclear how IIT attempts to quantify the integrated information for complex dynamical systems, given that previously the author stated that IIT can only be run on small systems.

- We changed patterns for causal structured in a couple of places.

- With the new structure of the introcuction (see below) we don’t introduce the issues with the system size until later, thus we hope this does not sound confusing anymore.

In IIT 3.0 [1], the lastest version of the framework proposed by the lab of Giulio Tononi

Spelling mistake. Additionally, I’m not sure this is the best way to reference the paper.

- Changed to “the latest version of the framework of Integrated Information Theory”

can only be computed for binary networks composed of up to a dozen of units. A variety of other measures exist to compute integrated information [3] and some of them are computationally lighter, but all share these limits to some extent.

What about all the metrics and adaptations that are used for brain imaging? Are those for binary networks as well? In what sense are large systems still a problem for the existing heuristics? There is not enough background to motivate these issues here.

- We now mention explicitely these versions for continuous distributions. To our knowledge, these versions of Phi are applied assuming Gaussian distributions of the data. In this new version we mention that they exist and are a way to compute integration analytically (therefore avoiding computation costs) and explain we do not consider Gaussian assuptions in this paper and focus on the computation of phi for general cases (lines 56-59).

Since most versions of Φ require computing distance measures between probability distributions of a system and finding the minimum information partition (MIP), they present important restrictions in terms of computational cost as a system scales in size.

Is the restriction really because of differences between probability distributions? Isn’t the MIP by far the limiting factor, given it is a bell number?

- The size of the computation of the MIP scales faster with size, but the computation of distances between probability distributions is still an important problem (e.g. for Ising models of size 20-30 it would already be difficult to calculate in a standard computer). If the reviewer feels that we should clarify this further we are open to suggestions.

Previous work [4] has tried to analyse how integrated information changes with spatial and temporal coarse graining in small networks

Did they try and fail? Grammar.

- We have rephrased this to remove the “try”

In general, measures of Φ can only be applied to small networks

What about all the ones for brain imaging. They are just ignored consistently throughout the motivating section of the text.

- We think that our comment above also applies to this.

In the framework of IIT 3.0, it is required that a system satisfies the Markov property (i.e., the state at time t only depends on the state at time t − τ)

In Hoel et al. 2016 “Can the macro beat the micro? Integrated information across spatiotemporal scales” there are clearly systems that are second-order

- I’m afraid I do not understand what the reviewer means here or what should be corrected. Could the reviewer clarify this?

we define a general model defining causal temporal interactions between variables

What would a causal non-temporal interaction be?

- We removed “temporal”

in general the MIP still is one that isolates one unit of one of the regions comprised by the mechanism.

Where is this shown “in general”

- We removed “in general” and tried to make this paragraph clearer.

the level of irreducibility of the causal distributions

What are causal distributions?

- Changed to “causal structures”

As IIT 3.0 operates with the transition probability matrix of a system, one could compute this matrix from time t to time t + τ and compute a new probability transition matrix for a bipartition by injecting noise in the connections affected by it at time t. This implies that the system behaves normally for the following steps.

It’s unclear what’s going on here. More time needs to be explicitly spent saying why this is different, perhaps with an actual example figure if this difference is a key point of the paper.

- We have rewritten parts of this section trying to make clearer the distinction between what we call “initial noise injection” and “continuous noise injection” for computing partitions. We hope the new version is better explained, but we are open to suggestions of further changes

The figure text from this figure is just lifted from the figure text of the previous figure. And there’s no figured 3 even cited in the text. And B in this “Figure 3” looks a lot the C in Figure 2, but cropped differently.

- This was a mistake, the legend has been corrected.

- Yes, the two figures are identical since, at is it explained in the text, the value of phi is equivalent in these two cases. We have reference the two figures together in the text for clarity.

This behaviour is completely different to the one from in the effect repertoire. This should not happen for a homogeneous system in a stationary state as the one under study here.

Grammar. Also, how is the reader to know this isn’t due to your modifactions in the calculation of integrated information to adapt to Ising models rather than if one actually calculate using the real values from IIT 3.0, even though it is computationally intractable?

- We modified the text to point that this more of a intuition, given the homogeneity of the system, and placed more weight in the fact phi_cause fails in this case to characterize the critical point as the point with more integrated information.

- Still, we think that our example shows the effects of using a uniform prior even for tau=1, which should be equivalent to the results in IIT3.0 (since our assumptions involve mostly the cases when tau>1). If the reviewer thinks that this should be discussed further we are open to the possiblity

Also, again the measure of φcause fails to capture integration around the critical point.

What’s the argument behind this being necessary?

- Outside the critical point, in the ordered and disordered regions, the fluctuations of the energy and the correlations of an Ising model are very small, and they tend to zero as the system grows in size. Thus, the only point where integrated information can be significantly high is around the critical point. This however was not clear in the previous version of the text. We now mention this more clearly in the introduction section and at the beginning of the results section.

These results illustrate that different distance metrics can have important effects in the behaviour of φ.

This is an interesting result and should be highlighted a bit more.

- We have extended the discussion of these results in section 3.3 and highligted the importance for debates in IIT.

We measure the integrated information of the mechanism φM under three different assumptions: a) that units outside of the mechanism operate normally, b) units outside the mechanism are independent noise sources, and c) units outside the mechanism are fixed as external constraints.

Again, this is quite interesting. Both the notion of exploring the difference between distributions and how to treat outside sources (as noise or so on) is extremely interesting. However, this is only explored in relationship to the Ising model. So the claims are somewhat incomplete, given the adaptations that are made. What about doing this is Markov chains and asking the same questions?

- The dynamical Ising model presented here would be an example of a Markov chain. The results of the paper could be reproduced in other Markov chains as long as they had the same homogeneous distribution of coupling parameters as the Ising model tested, but then the models would be very similar to the Ising models in the paper.

- I’m afraid that it is difficult to reproduce the results here in more general Markov chains with heterogeneous parameters (at least without any kind of approximation).

- These questions could be asked in general for small Markov chains, but I guess that would be a different paper. If the reviewer has a more specific suggestion we’d be happy to consider it.

GENERAL THOUGHTS 
The paper begins by arguing that IIT 3.0 doesn't scale in terms of its calculation and then introduces all sorts of assumptions in the Ising model to make it calculable. This doesn't actually solve the scaling problem so much as show that as long as enough assumptions are made, some version of IIT can be calculated. The paper begins by motivating itself by talking about scale but the takeaway is actually that assumptions in the calculation of IIT, specifically in terms of the chosen difference method and then also the choice of what distribution to use for elements outside the system, can be supported or disproven by watching how the measure operates in the Ising model. These are two mostly different problems, and yet the paper is motivated by the one that's not solved or really even argued for in the paper itself.

- We believe this point is very important, and we have tried to rearrange the whole introduction section in the line suggested by the reviewer. Now, we first introduce the problem of evaluating assumptions about how IIT can be defined, and only then introduce the issue of size as a particular problem that we find when we thing about testing IIT in models of critical phase transitions. We hope that this new structure is better for motivating the problem.

Overall, my impression of this paper is that it is a draft. There are numerous mistakes throughout involving grammatical errors, misplaced figures and figure texts. Two different times the figure texts of two different figures are identical. There is a figure included that is never cited. This is not a polished or final paper and should not have been submitted as such. There is a lot of repetition, and many sections could be a lot shorter. That said, some of the results are quite interesting and the general idea of the paper is based on an interesting argument – that PHI needs to be computed in a contextual manner dependent on the system. I do not believe it should be published in its current condition, but is due for a major revision that involves polishing and editing. In general the paper needs a stronger motivation, with the conclusions driving the original motivation.

- We thank the reviewer for the many corrections and suggestions. Given the short time given for returning the review, we performed important changes in some sections and we hope we have brought the paper to closer to a final form. Still, we will further proofread the manuscript to polish the current version.

p { margin-bottom: 0.1in; line-height: 115%; }

Reviewer 2 Report

In the present paper, the authors analyze the limitations of different versions of Phi, and they propose ways to overcome such limitations. However, talking about different versions of Phi is misleading since they blend together Phi 3.0, which can be computed over binary networks composed up to a dozen of units, and approximations of Phi computed over continuous and real-valued time series. For this reason, I would suggest improving the first part of the introduction by specifying which version of Phi they are questioning on a case-by-case basis.

The mean-field approximation presented in (Aguilera et al., Neural Networks, 2019) shows integration divergence when J approaches the critical point from the right. On the contrary, the level of integration is zero on the left of the critical point. Is it correct looking for a phi measure able to capture integration as the system approaches the critical transition from any side? Could the author expand this point and elaborate more about the relationship between the mean-field and the finite-size networks in designing this work?

I would also ask to clarify the concept explained at lines 193-194 and 322-326.

Minor: 

-      Legend in Figure 2 is unprecise: B and C do not describe the corresponding graphs;

-      Legend in Figure 4 in incomplete (it actually misses the description of Fig.4 B)

-      Line 118: “…mechanism integration of a mechanism…”, could the author resphrase this sentence?

Typos:

There are many typos in the text. For instance,

-      Line 14: the problem consciousness

-      Line 39: paractical difficulties

-      Line 47: dynamics its complex

-      Line 94: {s_i(t+1)}\in P, instead of \in M

-      Line 124: we divide system is divided into

-      Line 136: MIP in most cases one, verb missing

-      Line 156: There is also some aspects

-      Etc.

-      Line 247: It is well that

Author Response

In the present paper, the authors analyze the limitations of different versions of Phi, and they propose ways to overcome such limitations. However, talking about different versions of Phi is misleading since they blend together Phi 3.0, which can be computed over binary networks composed up to a dozen of units, and approximations of Phi computed over continuous and real-valued time series. For this reason, I would suggest improving the first part of the introduction by specifying which version of Phi they are questioning on a case-by-case basis.

We have rearranged the introduction to make the claims of the paper more clear. As well, we have tried to differentiate between IIT and its versions of Phi (as a theory of consciousness) and different versions of Phi (as general complexity integration measures). From the latter, we explicit mention Gaussian approximations that are used by some authors to avoid some of the computational costs by computing the values of phi analytically, and say that our work doesn’t consider these approximations. We hope these new version is less confused, but we could make the disctintion more explicit if the reviewer feels that is necessary.

The mean-field approximation presented in (Aguilera et al., Neural Networks, 2019) shows integration divergence when J approaches the critical point from the right. On the contrary, the level of integration is zero on the left of the critical point. Is it correct looking for a phi measure able to capture integration as the system approaches the critical transition from any side? Could the author expand this point and elaborate more about the relationship between the mean-field and the finite-size networks in designing this work?

Respect integration is zero when approaching a critical point from the disorder side, we have introduced a clarification in section 3.1. There we explain that the fact that integration is zero in the disordered side for the infinite system and not in the finite system is not a problem of the measure but a characteristic of the system. In the infinite size system units have independent dynamics until J reaches the threshold of the critical point. For finite size, units are not completely independent and our measure correctly captures non-zero integration.

As well, in section 3.3 we extended the discussion about the relation of the results in finite size and the infinite mean-field case. Specially, it is interesting that the Wasserstein distance has the same behaviour in finite and infinite cases, while the Kullback-Leibler divergence does not.

I would also ask to clarify the concept explained at lines 193-194 and 322-326.

In 193-194 we have tried to clarify the difference between what we called initial and continuous noise injections. We hope the new version is clearer respect to what we did in each case.

We also extended lines 322-326 to describe better what we did

Minor:

-      Legend in Figure 2 is unprecise: B and C do not describe the corresponding graphs;

-      Legend in Figure 4 in incomplete (it actually misses the description of Fig.4 B)

-      Line 118: “…mechanism integration of a mechanism…”, could the author rephrase this sentence?

These error have been corrected

Typos:

There are many typos in the text. For instance,

-      Line 14: the problem consciousness

-      Line 39: paractical difficulties

-      Line 47: dynamics its complex

-      Line 94: {s_i(t+1)}\in P, instead of \in M

-      Line 124: we divide system is divided into

-      Line 136: MIP in most cases one, verb missing

-      Line 156: There is also some aspects

-      Etc.

-      Line 247: It is well that

We have corrected these and other typos.

p { margin-bottom: 0.1in; line-height: 115%; }

Reviewer 3 Report

In this paper, the author critically reviews the assumptions IIT3.0 uses to calculate integrated information using a homogeneous kinetic Ising model. I believe that doing IIT’s analyses using the Ising model provides valuable information about how integrated information behaves. This work is relevant because it shows some useful properties that can massively reduce computational cost. On the other hand, there are some parts I had trouble with understanding. So, I recommend the author to address my concerns below:

(1) In figure 2 and figure 3, the author used ‘continuous’ and ‘initial’ injection of noise. Although these different ways of noise injections cause qualitatively distinct behaviors of integrated information, I could not find the description of the difference between them. Please clarify this.

(2) The author examines how phi value changes with the time-lag used to calculate phi, which is important to find the best time-lag that maximizes phi value according to the exclusion axiom of IIT. However, I could not understand his following point: ‘As we increase tau, this peak moves towards the critical point. Also, as tau increases the size of the peak decreases, tending to zero. This is a problem since phi(tau) is not able to capture maximum integration at the critical point.’ I understand that it’s natural to expect that phi would be maxed at the critical temperature. However, I’m not sure that it’s generally true that infinite time-lag gives the max phi. Rather, I expect to see a decrease of phi with time-lag, because the causality between mechanism and purview should be diluted with it. Please explain the necessity to take the infinite for the time-lag.

(3) The author argued that the behavior of cause phi should be similar to effect phi because the different behavior ‘should not happen for a homogeneous system in a stationary state as the one under study here’. However, I could not find any description of the reason the author thought so. Please clarify it.

(4) The author found weird behavior of phi when he/she injected independent noise to calculate phi (Figure 5B) in order to marginalize out units outside the mechanism. However, the standard way of IIT’s marginalization would be marginalizing the probability distribution with the uniform joint distribution. Please explain why the author did not do this and show what happens if it is done.

(5) I cannot see how the author constructs the system shown in figure 6A. What is the local connection in A and E? Also, although the author described that ‘one region A with self-interaction and another region E which is just coupled to the first without recurrent connections,’ it looks like there are bidirectional connections between A and E. Also, the legend of figure 6 does not make sense at all. It’s important to correctly understand results shown figure 6 so please fully revise relevant parts.

Minor points

(1) The legend is hard to understand. It seems the difference between figures 2A and 2B is the difference in how to inject noise, but the label of y says it is cumulative phi instead of actual phi.

(2) There is a typo in line 195: ‘integrated information integration’

(3) The legend of figure 3 looks identical to that of figure 2. Please revise it.

(4) A space between ‘vice’ and ‘versa’ is missed in line 211.

(5) ‘Figure 2.A’ in line 213 might be ‘Figure 3A’.

(6) ‘Figure 2.B’ in line 222 might be ‘Figure 3B’.

(7) The legend of figure 4 does not make sense.

Author Response

In this paper, the author critically reviews the assumptions IIT3.0 uses to calculate integrated information using a homogeneous kinetic Ising model. I believe that doing IIT’s analyses using the Ising model provides valuable information about how integrated information behaves. This work is relevant because it shows some useful properties that can massively reduce computational cost. On the other hand, there are some parts I had trouble with understanding. So, I recommend the author to address my concerns below:

We thank the reviewer by these comments. We have tried to addressed all the points raised, which we comment below each paragraph. As well, the introduction has been rearranged in response to comments from the other reviewers.

(1) In figure 2 and figure 3, the author used ‘continuous’ and ‘initial’ injection of noise. Although these different ways of noise injections cause qualitatively distinct behaviors of integrated information, I could not find the description of the difference between them. Please clarify this.

This comment is right, the terms were described very superficially. We have tried to define  both terms and clarify the difference between both in section 3.1 .

(2) The author examines how phi value changes with the time-lag used to calculate phi, which is important to find the best time-lag that maximizes phi value according to the exclusion axiom of IIT. However, I could not understand his following point: ‘As we increase tau, this peak moves towards the critical point. Also, as tau increases the size of the peak decreases, tending to zero. This is a problem since phi(tau) is not able to capture maximum integration at the critical point.’ I understand that it’s natural to expect that phi would be maxed at the critical temperature. However, I’m not sure that it’s generally true that infinite time-lag gives the max phi. Rather, I expect to see a decrease of phi with time-lag, because the causality between mechanism and purview should be diluted with it. Please explain the necessity to take the infinite for the time-lag.

This comment reveals a problem that is interesting, but not easy to solve. The main difficulty is that systems at critical points tend to display infinite correlation lengths and critical slowing down, thus they dynamics can only be completely captured with infinite time-lags

With the ‘initial noise injection’ version of partitions, it is hard to capture long timescales of critical system, and phi is maximized for points at the ordered side of the phase transition . With the ‘continuous time injection’ , these large time-scales are captured. The price to pay for that, is that now the measure measures the influence of different timescales cummulatively, and thus there is no longer a dilution of causal influences with larger lags.

We have included this discussion in section 3.1 and discuss why we propose that the second approach is more adequate for the examples covered in the paper.

(3) The author argued that the behavior of cause phi should be similar to effect phi because the different behavior ‘should not happen for a homogeneous system in a stationary state as the one under study here’. However, I could not find any description of the reason the author thought so. Please clarify it.

We have clarified in the text that this claim is just an intuition of what should happen, and we have placed more importance in the fact that in this case the integration does not match with the description of the phase space, displaying very little integrated information at the critical point and large integrated information deep into the ordered phase.

(4) The author found weird behavior of phi when he/she injected independent noise to calculate phi (Figure 5B) in order to marginalize out units outside the mechanism. However, the standard way of IIT’s marginalization would be marginalizing the probability distribution with the uniform joint distribution. Please explain why the author did not do this and show what happens if it is done.

I am not sure if I understand this problem correctly. In IIT 3.0 uniform independent noise is injected to units to create a uniform distribution, and this is what we did. Other versions of IIT may involve marginalization over probability distributions in the same way that computing mutual information, but that is the difference between traditional information theoretic approaches and the ‘interventional’ or ‘perturbational’ approach used by IIT 3.0 (along with Pearl, and others like Ay and Polani). I’m not sure if the question of the reviewer is related to this, but we can clarify it in the text if necessary.

(5) I cannot see how the author constructs the system shown in figure 6A. What is the local connection in A and E? Also, although the author described that ‘one region A with self-interaction and another region E which is just coupled to the first without recurrent connections,’ it looks like there are bidirectional connections between A and E. Also, the legend of figure 6 does not make sense at all. It’s important to correctly understand results shown figure 6 so please fully revise relevant parts.

There was a mistake in the legend of Figure 6, it is corrected now. We have also tried to clarify the description of the system connectivity in the text (section 3.5)

Minor points

(1) The legend is hard to understand. It seems the difference between figures 2A and 2B is the difference in how to inject noise, but the label of y says it is cumulative phi instead of actual phi.

(2) There is a typo in line 195: ‘integrated information integration’

(3) The legend of figure 3 looks identical to that of figure 2. Please revise it.

(4) A space between ‘vice’ and ‘versa’ is missed in line 211.

(5) ‘Figure 2.A’ in line 213 might be ‘Figure 3A’.

(6) ‘Figure 2.B’ in line 222 might be ‘Figure 3B’.

(7) The legend of figure 4 does not make sense.

Thanks for the corrections. We have incorporated all in the text. We have also corrected the legend of figures 2 and 3.

Round 2

Reviewer 1 Report

Small changes

(IIT 3.0) are difficult to apply to systems larger than a dozen of  units

Grammar.

Due to combinatorial explosion, computing Φ in practice is only possible in small discrete  systems, preventing its application to the very large or even infinite system where critical dynamics can be appreciated

English

Moreover, since we will assume a homogeneous architecture, mechanisms in the system have a similar behaviour in most cases, and the distinction between mechanism integration φ and system-level integration Φ is not quite revealing (as we will find diverging values of φ at some points, this will correspond to equally diverging values of Φ). Thus, for simplicity, we will compute only the integrated information of a mechanism, φ, comprising the whole system of interest

We need something more than the authors word on this, particularly since “big PHI” may explain some of the discrepancies that the paper points out. Why can “big PHI” or system-level integration be completely ignored in this account? We never see any evidence of this, even just something showing how it grows or obeys the same rules as the “small PHI” directly. Is it really true that the values are identical, or follow immensely simply from the “small PHI” calculation in this system? If true, this seems a point of interest given the new direction of the paper of discussing how the assumptions behind IIT depend on the system. I would suggest incorporating this into the results in some way.

Figure 6E is not referenced in the figure text.

Second, some of the assumptions made by IIT 3.0 for computing integration over the cause repertoire of a state (the distribution of previous states) present problems for capturing the integration  of a system at a critical point. The assumption of a uniform prior distribution simplifies the calculations of cause repertoires but distorts the behaviour of the system.

In the section of the cause information it is not taken into account that the minimum bit value is taken between the cause and effect information within IIT 3.0. This should be compared in the paper to draw this conclusion. In general the conclusions of the piece are a bit stronger than the given evidence provided, since IIT is generally only calculated in parts. I’m not sure why these weren’t done, other than ease of calculation.

In contrast, IIT 3.0 quantifies the level of integration dismissing the  interaction between a system and its environment.

Grammar

Specifically, IIT 3.0 considers the units outside of a system as static variables and the units within a system but outside of a mechanisms as independent sources of noise.

Grammar

This allows to draw important consequences for the application of integrated information measures in simulated and experimental setups.

Grammar

Overall comments and further recommendations

Overall, this is an interesting paper with a lot to say. It is much easier to understand now that problems with presentation and image/text duplication are resolved and the structure is much clearer, as well as its reasoning and its final conclusions. I do think the author’s work has a lot of merit and that there are some great contributions to the literature here, particularly for people looking to design future versions of IIT and also apply IIT to different systems. However, the remaining problem that I see is that the paper takes leaps in a few places, but does not back up those leaps with its simulations. This undermines the strength of the conclusions by just leaving out various simulations that could be run. For example, this paper never once calculates the “big PHI” metric, which is the main metric of IIT 3.0, yet still takes all of its conclusions as applying directly to IIT 3.0. A similar problem is that IIT 3.0 takes the minimum between the cause/effect information. They are analyzed only separately here yet the conclusions are assumed to still apply directly to IIT 3.0. Given these two missing components I don’t think the author’s conclusions actually can be proven to follow from what is presented. I worry that the paper’s results will be dismissed simply because it does not account for the simple criticism of “the big PHI was never calculated and the cause/effect information were never treated in relation to each other. Therefore, it’s unknown how these results actually apply to the IIT 3.0 calculation, merely its parts.”

I suggest that the paper’s strong conclusions be backed up by actually examining these two values: a) the big PHI and b) the min(cause/effect) and their behavior in the model. It may not be necessary to calculate the “big PHI” for all the examples if, as the author claims, big PHI acts identically to small PHI, and this is actually shown in the paper rather than merely being assumed. Indeed, if big PHI is essentially unnecessary in the calculation of IIT in an Ising model, this also supports the author’s point that a sensible calculation of integrated information changes depending on the system under question. Otherwise the author needs to be constantly mentioning that their conclusions apply only to the “small PHI” calculation in IIT 3.0, and even that is not done 100% as in IIT 3.0 since the behavior of the minimum is never shown in this paper. Without these two comparisons the conclusions of the paper in their current form are presented far too strongly. However, once those two things are shown in the paper I’d strongly recommend it for publication as a solid contribution to the existing literature.

Author Response

We introduced the grammar corrections pointed by the reviewer

Figure 6E is not referenced in the figure text.
The figure (now Figure 7E) is referenced in the text together with Figure 7D (line 393)

We also thank the reviewer by their reflections about big PHI and the conclusions of the paper. Motivated by these comments we decided to apply the same methods we used to simplify computations in the system with just one region in order to compute big PHI.
We added a new section (Section 3.5) discussing the results of measuring big PHI in this simple system, showing that divergences at the critical point are conserved (and amplifying) and discussing about the necessity of adding this second level of integration over the values of small phi

About the min(cause/effect) metric, we think that some parts of the text addressing this might not be clear and we have rewritten them. First, when discussing the value of phi_cause in Figure 3.A, we describe now more clearly that its values are much lower than those of phi_effect (Figure 2.C), thus the values of phi_cause dominate phi.
As well, when we assume stationarity instead of assuming an independent prior distribution, phi_cause = phi_effect = phi, then it is not necessary to compute the minimum (and this is what we assume during the rest of the paper).
We also mention as a suggestion that min(phi_cause,phi_effect) should be computed in a case where instead of a stationary distribution, some other specific transient distribution is assumed, although in this case the computational complexity of the measure would increase.
We hope that these clarifications can address the concerns of the reviewer and that it can be clear for a reader that the conclusions are backed by the results (at least in the stationary case).
If the reviewer thinks that it is necessary to compute some instance of min(phi_cause,phi_effect) it might be done, though it would take some time to implement and perform new simulations and would also maybe adding a subsection explaining the methods and showing those results...

Round 3

Reviewer 1 Report

The comparisons of BIG PHI makes the paper much more complete. Indeed, BIG PHI does seem to have good respect for the critical point under certain assumptions (which the author rightly points out are need) - the discussion is still quite negative but I view this as actually somewhat supportive of IIT. I think it was always a straw-man to assume that a measure designed to work in discrete finite systems of logic gates would always work exactly the same across all system types (like continuous systems, physically-realistic systems, etc). However, the author has shown that with certain (sensible) assumptions IIT does indeed capture the critical point, there just needs to be thought in on what those assumptions are on a case-by-case basis. So the author could land on a slightly more positive note that currently. 

Please do a last grammatical check.

Author Response

Many thanks for your comments.

We reviewed the text and corrected a couple of typos. We will perform a thorough revision before production.

As for the discussion, we have added a new paragraph inspired by your comment. We hope that it can contribute to read the results of the paper as a constructive contributions to IIT and future versions of the theory.

"Since IIT~3.0 has been mostly tested in small logic gate circuits, exploring the behaviour of integrated information in large Ising models has allowed us to investigate questions that were so far unexplored and inspect some of the assumptions of the theory from a new perspective. We consider that the value of the study is twofold. On the one hand, we propose a family of models with known statistical properties, where calculations of integrated information are simplified. These and similar models could work as a benchmark for testing properties of integrated information in large systems. On the other hand, the reformulations of different aspects of the theory proposed during the paper could be considered by future versions of IIT in order to capture some of the phenomena that we could expect in large complex systems."